

# The prognostic significance of lymph nodes in patients with pT1c33N0M0 non-small cell lung cancer: a retrospective study

Wei Yang and  Luyi Wang

Department of Thoracic Surgery, Beijing Chest Hospital, Capital Medical University, Beijing, China
Rehabilitation Diagnosis and Treatment Center, Beijing Rehabilitation Hospital of Capital Medical University, Beijing, China

## ABSTRACT

**Objective**. The objective of this study was to appraise the prognostic impact of lymph nodes in patients diagnosed with pT1c33N0M0 non-small cell lung cancer (NSCLC) and to delve into the prognostic significance of lymph nodes located at the N1 lymph node station in this patient cohort.

**Methods**. A retrospective analysis of clinical data was conducted for 255 patients diagnosed with pT1c33N0M0 NSCLC. Lymph nodes were tabulated and categorized into three groups (0–10 nodes, 11–16 nodes, >16 nodes). Clinical data among these three groups of pT1c33N0M0 NSCLC patients were compared. We conducted both univariate and multivariate analyses to pinpoint the factors that impact the prognosis of patients with pT1c33N0M0 non-small cell lung cancer (NSCLC). Additionally, we employed receiver operating characteristic (ROC) curve analysis to pinpoint the optimal lymph node criteria at the N1 station for prognostic prediction in pT1c33N0M0 NSCLC patients.

**Results**. Within the cohort of 255 individuals afflicted with pT1c33N0M0 non-small cell lung cancer (NSCLC), a comprehensive tally of 3,902 lymph nodes was diligently established, yielding an average of 15.3 nodes for each patient. Multivariate analysis demonstrated that tumor size, T stage, and lymph nodes were independent factors significantly impacting the prognosis of pT1c33N0M0 NSCLC patients ($P < 0.05$). ROC curve analysis revealed an area under the curve of 0.6982 for predicting prognosis using N1 station in pT1c33N0M0 NSCLC patients. The maximum Youden index was observed at an N1 station of 2.7 nodes. Patients with N1 station $\geq$ three nodes had significantly better prognoses compared to those with < 3 nodes (both $P < 0.05$).

**Conclusion**. Lymph nodes serve as an independent prognostic factor for pT1c33N0M0 NSCLC patients. Detecting at least three or more lymph nodes at the N1 station is associated with a more favourable prognosis in pT1c33N0M0 NSCLC patients.

Corresponding author
Luyi Wang, wangly992012@163.com

## INTRODUCTION

Lung cancer is a common malignancy in clinical practice, known for its high incidence and mortality rates (*Zhu et al., 2023*). Lung cancer is typically divided into two primary

categories: small cell lung cancer and non-small cell lung cancer (NSCLC). NSCLC is the more prevalent variant, constituting roughly 80% to 85% of diagnosed cases (*Sung et al., 2021*). NSCLC is highly malignant and is influenced by various factors, including genetics, environmental factors, and smoking. Clinical manifestations often include chest pain, coughing, and low-grade fever, among others. Early-stage NSCLC is typically asymptomatic, leading to late-stage diagnoses with a 5-year survival rate of less than 20% (*Martinez-Zayas et al., 2021*). Current clinical management of NSCLC often involves surgical removal of the affected lung lobe followed by comprehensive mediastinal lymph node dissection. For patients with lymph node involvement, adjuvant chemotherapy is usually necessary to reduce the risk of recurrence. The accuracy of lymph node detection is pivotal in tailoring treatment strategies (*Pechoux et al., 2022*). Lymphatic metastasis is an important way for cancer cells to metastasize, so lymph node dissection is of great clinical significance for cancer patients to prevent cancer cell metastasis.

Lymph nodes play a significant role in pathological staging and prognosis assessment, as examining an appropriate number of lymph nodes enhances the reliability of staging and the precision of prognosis evaluation (*Shaw et al., 2019*). However, there is no established consensus on the optimal number of lymph nodes to be removed during surgery for pT1c33N0M0 NSCLC patients. Furthermore, most research has concentrated on the number of mediastinal lymph nodes, with limited investigation into lymph nodes at the N1 lymph nodes station (stations 10-14). Because N1 station are located deeper anatomically, they are often challenging to detect, potentially resulting in false-negative results and reduced opportunities for postoperative adjuvant therapy, consequently increasing the risk of postoperative recurrence. Henceforth, this study conducts a retrospective examination of clinical data pertaining to 255 patients diagnosed with pT1c33N0M0 NSCLC who underwent curative lung cancer surgery. The primary goal is to investigate the correlation between various lymph nodes and the most advantageous quantity of N1 station lymph node dissections concerning the prognosis of pT1c33N0M0 NSCLC patients, with the ultimate aim of furnishing dependable guidance for clinical treatment decisions.

## MATERIALS AND METHODS

### Data

We undertook a retrospective examination of 255 individuals diagnosed with pT1c33N0M0 NSCLC who received treatment at Beijing Chest Hospital, Capital Medical University, spanning from January 2020 to July 2022. The eligibility criteria were established as follows: (1) histopathologically confirmed NSCLC with a pathological stage of pT1c33N0M0, (2) patients who underwent systematic mediastinal lymph node dissection, and (3) availability of complete medical records. Exclusion criteria included: (1) patients with severe cardiopulmonary insufficiency, (2) preoperative neoadjuvant therapy, (3) sublobar resection or the absence of systematic lymph node dissection, (4) intraoperative discovery of thoracic or distant metastasis, and (5) incomplete medical records. Within the group of 255 patients diagnosed with pT1c33N0M0 NSCLC, 169 were male, and 86 were female, spanning an age range of 33 to 81 years. A collective count of 3,902 lymph nodes was

identified, equating to an average of 15.3 lymph nodes per patient. Lymph nodes were divided into three groups based on quartiles: <10 nodes, 11–16 nodes, and >16 nodes, as there were no specific grouping criteria for lymph nodes. All samples obtained in this study were approved by the ethics committee of the Beijing Chest Hospital, Capital Medical University and abided by the ethical guidelines of the Declaration of Helsinki, and ethics committee agreed to waive informed consent.

## Methods
### Clinical data
Clinical data, including patient gender, age, smoking history, surgical site, maximum tumor diameter, T stage, pathological type, and preoperative comorbidities, were obtained from the electronic medical record system. Lymph nodes of lung cancer were detected by diffusion weighted magnetic resonance imaging. The latest eighth edition of TNM staging of lung cancer issued by the International Association for the Study of Lung Cancer still follows the seventh edition of the N staging method, and N1 is defined as ipsilateral peribronchial and/or ipsilateral hilar lymph nodes and intrapulmonary lymph nodes with metastasis, including direct invasion of the primary tumor (*Asamura et al., 2015*). Follow-up was conducted once every 3 months in the first year, once every 6 months in the second year, and once annually from the third year onwards.

### Outcome measures
Patient outcomes were assessed using follow-up data obtained from patient re-examinations or readmissions. The 5-year survival rate is an important index to evaluate the effect of tumor treatment. The primary outcome measure was the 5-year survival rate, calculated as follows: (number of cases alive at the end of n years of follow-up/number of cases at the beginning of follow-up) * 100%. The secondary outcome was median survival time, defined as the survival time corresponding to a survival function value of 0.5.

## Statistical methods
Data were analyzed using SPSS 25.0 software (SPSS Inc., Chicago, IL, USA). Chi-square tests were used to compare clinical and pathological data among multiple patient groups. Kaplan–Meier analysis was employed to calculate survival rates, and differences between groups were assessed using the Log-rank test, including trend tests. Variables with $P < 0.3$ in univariate analysis were included in a Cox proportional hazards model for multivariate survival analysis. The receiver operating characteristic (ROC) curve analysis was employed to establish the ideal cutoff value for prognostic predictions in pT1c33N0M0 NSCLC patients, focusing on the N1 station. Statistically significant distinctions were recognized ($P < 0.3$).

# RESULTS

## Lymph node detection and clinical data comparison among pT1c33N0M0 NSCLC patients with different lymph nodes
In the cohort of 255 patients with pT1c33N0M0 NSCLC, a collective sum of 3,902 lymph nodes was identified, resulting in an average of 15.3 lymph nodes per patient. Subsequently,

**Table 1  Comparison of clinical data among pT1c33N0M0 NSCLC patients with different lymph nodes.**

| Characteristics | Lymph nodes | | | $\chi^2$ | P |
|---|---|---|---|---|---|
| | 0–10 (n = 60) | 11~16 (n = 89) | >16 (n = 106) | | |
| Gender | | | | | |
| Male | 43 (71.67%) | 63 (70.79%) | 73 (68.87%) | 0.166 | 0.920 |
| Female | 17 (28.33%) | 26 (29.21%) | 33 (31.13%) | | |
| Age (years) | | | | | |
| ≤60 | 19 (31.67%) | 37 (41.57%) | 29 (27.36%) | 4.497 | 0.106 |
| >60 | 41 (68.33%) | 52 (58.43%) | 77 (72.64%) | | |
| Smoking history | | | | | |
| Yes | 16 (26.67%) | 33 (37.08%) | 27 (25.47%) | 3.485 | 0.175 |
| No | 44 (73.33%) | 56 (62.92%) | 79 (74.53%) | | |
| Surgical site | | | | | |
| Right upper lung | 12 (20.00%) | 17 (19.10%) | 23 (21.70%) | | |
| Right middle lung | 5 (8.33%) | 7 (7.87%) | 10 (9.43%) | | |
| Right lower lung | 9 (15.00%) | 9 (10.11%) | 14 (13.21%) | 1.795 | 0.987 |
| Left upper lung | 23 (38.34%) | 37 (41.57%) | 37 (34.91%) | | |
| Left lower lung | 11 (18.33%) | 19 (21.35%) | 22 (20.75%) | | |
| Tumor max diameter | | | | | |
| ≤3 cm | 41 (68.33%) | 68 (76.40%) | 54 (50.94%) | 14.262 | 0.001 |
| >3 cm | 19 (31.67%) | 21 (23.60%) | 52 (49.06%) | | |
| T stage | | | | | |
| T1c stage | 43 (71.67%) | 57 (64.04%) | 53 (50.00%) | | |
| T2 stage | 11 (18.33%) | 25 (28.09%) | 41 (38.68%) | 9.416 | 0.052 |
| T3 stage | 6 (10.00%) | 7 (7.87%) | 12 (11.32%) | | |
| Pathological type | | | | | |
| Adenocarcinoma | 44 (73.33%) | 56 (62.92%) | 54 (50.94%) | 12.116 | 0.017 |
| Squamous cell carcinoma | 14 (23.34%) | 22 (24.72%) | 43 (40.57%) | | |
| Other | 2 (3.33%) | 11 (12.36%) | 9 (8.49%) | | |
| Preoperative comorbidities | | | | | |
| Yes | 13 (21.67%) | 17 (19.10%) | 22 (20.75%) | 0.160 | 0.923 |
| No | 47 (78.33%) | 72 (80.90%) | 84 (79.25%) | | |

patients were categorized into three groups based on the number of lymph nodes detected. The first group had a lymph nodes of 0–10 nodes (60 cases, 23.5%), the second group had a lymph nodes of 11–16 nodes (89 cases, 34.9%), and the third group had a lymph nodes of >16 nodes (106 cases, 41.6%).

In terms of patient gender, age, smoking history, surgical site, T stage, and preoperative comorbidities, there were no statistically significant distinctions observed among the three groups ($P > 0.05$). However, significant differences were observed in tumour maximum diameter and pathological type distribution ($P < 0.05$), as detailed in Table 1.

### Univariate and multivariate analysis of factors affecting the 5-year survival rate of pT1c33N0M0 NSCLC patients

The outcomes of the univariate analysis revealed significant associations between the 5-year survival rate of pT1c33N0M0 NSCLC patients and variables such as tumour maximum diameter, T stage, and lymph nodes ($P < 0.05$, refer to Table 2). Those variables with $P$-values less than 0.3 in the univariate analysis were subsequently included in a Cox regression model for multivariate analysis. The results from this multivariate analysis indicated that tumour maximum diameter, T stage, and lymph nodes remained as independent factors significantly influencing the 5-year survival rate of pT1c33N0M0 NSCLC patients ($P < 0.05$, refer to Table 3).

### Impact of univariate and multivariate analysis on median survival time in pT1c33N0M0 NSCLC patients

The results of univariate analysis revealed a significant association between tumor maximum diameter, T stage, lymph nodes, and median survival time in pT1c33N0M0 NSCLC patients ($P < 0.05$, as shown in Table 4). Subsequently, variables with $P < 0.3$ in the univariate analysis were included in a Cox regression model for multivariate analysis. The results derived from the multivariate analysis demonstrated that tumor maximum diameter, T stage, and lymph nodes had independent impacts on the median survival time of pT1c33N0M0 NSCLC patients, with statistical significance ($P < 0.05$, as detailed in Table 5).

### Determination of the optimal number of N1 station detected for pT1c33N0M0 NSCLC patients

The ROC curve analysis revealed an area under the curve (AUC) of 0.6982 for predicting the 5-year survival of pT1c33N0M0 NSCLC patients based on the number of N1 station lymph nodes detected. The maximum Youden index, approximately 0.316, occurred when the number of N1 station detected was 2.7 nodes (see Fig. 1). Using a threshold of 2.7 nodes for the number of N1 station detected, the sensitivity for predicting 5-year survival was 62.7%, and the specificity was 68.9%. Consequently, rounding to the nearest whole number, a threshold of three nodes for the number of N1 station detected was chosen. Patients were categorized into two groups: those with less than three nodes detected in the N1 station (105 cases) and those with three or more nodes detected (150 cases). A comparison of the 5-year survival rate ($\chi^2 = 4.792$, $P = 0.027$) and median survival time ($\chi^2 = 4.236$, $P = 0.031$) revealed significant differences between the two groups.

## DISCUSSION

Non-small cell lung cancer (NSCLC) is a malignant tumor originating from the bronchial mucosa, bronchial glands, and alveolar epithelium. In recent years, the increasing prevalence of various carcinogenic factors has led to a rising incidence of this disease (*Spurr et al., 2022*). The pathogenesis of NSCLC is complex and not yet fully understood clinically, but it is often associated with factors such as age, genetics, and smoking (*Jiang, Zhou & Huang, 2021*; *Saxena, 2023*). Early-stage NSCLC typically lacks specific symptoms,

**Table 2** Univariate analysis of factors affecting the 5-year survival rate of pT1c33N0M0 NSCLC patients.

| Characteristics | N (%) | 5-year survival rate (%) | $\chi^2$ | P |
|---|---|---|---|---|
| Gender | | | | |
| Male | 169 (66.27%) | 56.24% | 0.123 | 0.784 |
| Female | 86 (33.73%) | 55.71% | | |
| Age (years) | | | | |
| ≤60 | 85 (33.33%) | 58.74% | 0.127 | 0.790 |
| >60 | 170 (66.67%) | 52.58% | | |
| Smoking history | | | | |
| Yes | 76 (29.80%) | 52.87% | 0.040 | 0.891 |
| No | 179 (70.20%) | 54.79% | | |
| Surgical site | | | | |
| Right upper lung | 52 (20.39%) | 49.17% | | |
| Right middle lung | 22 (8.63%) | 40.73% | | |
| Right lower lung | 32 (12.55%) | 56.77% | 0.067 | 0.999 |
| Left upper lung | 97 (38.04%) | 53.28% | | |
| Left lower lung | 52 (20.39%) | 56.72% | | |
| Tumor max diameter | | | | |
| ≤3 cm | 163 (63.92%) | 66.92% | 21.056 | <0.001 |
| >3 cm | 92 (36.08%) | 38.76% | | |
| T stage | | | | |
| T1c stage | 153 (60.00%) | 66.39% | | |
| T2 stage | 77 (30.20%) | 45.34% | 29.913 | <0.001 |
| T3 stage | 25 (9.80%) | 10.48% | | |
| Pathological type | | | | |
| Adenocarcinoma | 154 (60.39%) | 54.71% | | |
| Squamous cell carcinoma | 79 (30.98%) | 52.33% | 0.276 | 0.871 |
| Other | 22 (8.63%) | 68.87% | | |
| Preoperative comorbidities | | | | |
| Yes | 52 (20.39%) | 40.58% | 0.029 | 0.877 |
| No | 203 (79.61%) | 57.82% | | |
| Lymph nodes | | | | |
| 0–10 | 60 (23.53%) | 41.52% | | |
| 11–16 | 89 (34.90%) | 56.73% | 6.068 | 0.048 |
| >16 | 106 (41.57%) | 60.94% | | |

leading to delayed diagnosis and most patients being diagnosed at an advanced stage (*Guan et al., 2022*; *Chang et al., 2021*; *Grønberg et al., 2021*). The timely detection and early initiation of treatment play a pivotal role in enhancing the prognosis of patients with NSCLC.

Currently, the standard treatment for pT1c33N0M0 NSCLC patients involves pulmonary lobectomy with systematic mediastinal lymph node dissection, which effectively removes diseased tissue and prolongs survival. However, the problem of postoperative recurrence

**Table 3** Multivariate analysis of factors affecting the 5-year survival rate of pT1c33N0M0 NSCLC patients.

| Index | β | SE | Wald | OR (95% CI) |
|---|---|---|---|---|
| Lymph nodes | −0.334 | 0.037 | 81.523 | 0.716 (0.666, 0.770) |
| T stage | 3.798 | 0.475 | 7.893 | 3.798 (1.497, 9.636) |
| Tumor max diameter | −1.248 | 0.516 | 5.852 | 0.287 (0.104, 0.789) |
| Preoperative comorbidities | −0.265 | 0.196 | 1.832 | 0.767 (0.522, 1.126) |

**Table 4** Univariate analysis of factors affecting median survival time of pT1c33N0M0 NSCLC patients.

| Characteristics | N (%) | Median survival time (months) | $\chi^2$ | P |
|---|---|---|---|---|
| Gender | | | | |
| Male | 169 (66.27%) | 54.0 | 0.123 | 0.726 |
| Female | 86 (33.73%) | 52.0 | | |
| Age (years) | | | | |
| ≤60 | 85 (33.33%) | 56.0 | 0.127 | 0.722 |
| >60 | 170 (66.67%) | 52.0 | | |
| Smoking history | | | | |
| Yes | 76 (29.80%) | 53.0 | 0.040 | 0.842 |
| No | 179 (70.20%) | 54.0 | | |
| Surgical site | | | | |
| Right upper lung | 52 (20.39%) | 49.0 | 0.999 | 0.794 |
| Right middle lung | 22 (8.63%) | 41.0 | | |
| Right lower lung | 32 (12.55%) | 54.0 | 0.067 | 0.966 |
| Left upper lung | 97 (38.04%) | 54.0 | | |
| Left lower lung | 52 (20.39%) | 54.0 | | |
| Tumor max diameter | | | | |
| ≤3 cm | 163 (63.92%) | 64.0 | 21.056 | <0.001 |
| >3 cm | 92 (36.08%) | 40.0 | | |
| T stage | | | | |
| T1c | 153 (60.00%) | 64.0 | <0.001 | |
| T2 | 77 (30.20%) | 45.0 | 29.913 | |
| T3 | 25 (9.80%) | 10.0 | | |
| Pathological type | | | | |
| Adenocarcinoma | 154 (60.39%) | 54.0 | 0.871 | 0.647 |
| Squamous carcinoma | 79 (30.98%) | 52.0 | 0.276 | 0.871 |
| Other | 22 (8.63%) | 68.0 | | |
| Preoperative comorbidities | | | | |
| Yes | 52 (20.39%) | 40.0 | 0.029 | 0.865 |
| No | 203 (79.61%) | 57.0 | | |
| Lymph nodes | | | | |
| 0–10 | 60 (23.53%) | 41.0 | <0.001 | |
| 11–16 | 89 (34.90%) | 56.0 | 6.068 | |
| >16 | 106 (41.57%) | 60.0 | | |

**Table 5** Multivariate analysis of factors affecting median survival time of pT1c33N0M0 NSCLC patients.

| Index | β | SE | Wald value | HR (95% CI) |
|---|---|---|---|---|
| Tumor max diameter | −0.473 | 0.133 | 12.631 | 0.623 (0.450, 0.861) |
| T stage | 0.843 | 0.209 | 16.067 | 2.322 (1.448, 3.724) |
| Lymph nodes | −0.374 | 0.076 | 24.428 | 0.687 (0.580, 0.813) |

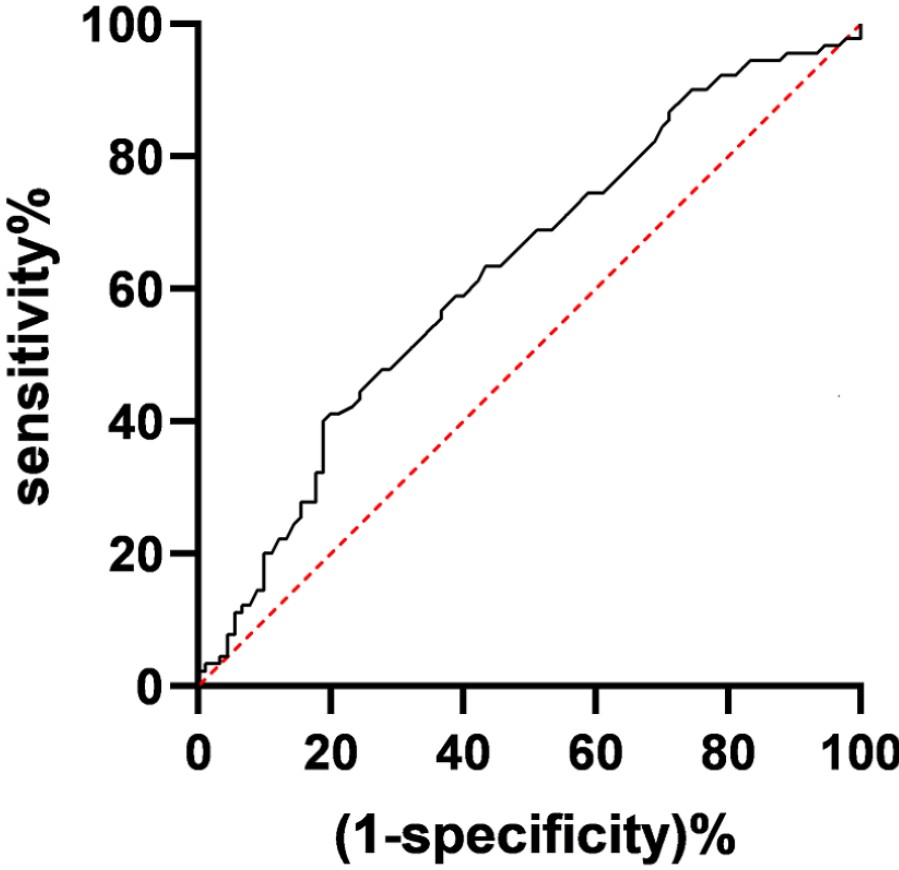

**Figure 1** Receiver operating characteristic (ROC) curve for the number of N1 station detected.

in pT1c33N0M0 NSCLC patients remains a challenge for clinical physicians (*Kamigaichi et al., 2020*). Clinical studies have suggested that the number of lymph nodes removed during surgery is one of the important factors affecting postoperative recurrence in these patients (*Raman et al., 2021*). Accurate lymph node detection plays a significant role in pathological staging and prognosis assessment for pT1c33N0M0 NSCLC patients. This study has revealed that the count of identified lymph nodes serves as an independent prognostic indicator for these patients. Furthermore, it was observed that a greater number of lymph nodes is correlated with a more favorable prognosis, aligning with findings from previous research (*Zhang et al., 2023*; *Feng et al., 2023*; *Gierada et al., 2022*).

This can be attributed to the fact that an increased number of lymph nodes detected can make lymph node metastasis more likely to be discovered. Micro-metastases in lymph nodes are often overlooked clinically. However, when the number of lymph nodes detected increases, it enables a more accurate assessment of patients originally classified as N0 to be reclassified as N1, ensuring that patients receive standardized postoperative treatment and thereby improving their survival rates (*Behinaein et al., 2023*). Furthermore, routine pathological examinations are typically performed by pathologists under a microscope, which may introduce subjectivity. Clinical pathologists have more opportunities to examine lymph node pathology sections, reducing the risk of diagnostic errors due to individual subjective experience (*Nounsi et al., 2023*). Thus, more accurate diagnosis is possible for pT1c-3N0M0 NSCLC patients, allowing for individualized postoperative adjuvant treatments that can significantly extend their survival and improve prognosis (*Katsumata et al., 2019*).

The results of this study also show that patients with ≥3 lymph nodes detected in the N1 station had better long-term survival rates compared to those with <3 lymph nodes detected. Therefore, it is beneficial for pT1c33N0M0 NSCLC patients to detect as many lymph nodes as possible. Surgeons should aim to detect as many N1 station as possible, including a layered dissection up to the 14th station lymph node, with a minimum detection count of ≥3 nodes. This approach leads to more accurate pathological staging after curative lung cancer surgery, which, in turn, provides patients with a better prognosis (*Maniwa et al., 2020*).

## CONCLUSIONS

The quantity of identified lymph nodes stands as an independent prognostic determinant for individuals diagnosed with pT1c33N0M0 NSCLC. A minimum of three or more lymph nodes should be detected in the N1 station for better prognosis in these patients. However, this study is a single-center retrospective analysis with certain case selection bias and a relatively small sample size. Moreover, the study did not conduct further analysis on specific T stages, which limits the generalizability of the results. Future research should consider conducting multicenter studies with larger samples and a multidimensional approach to further validate the impact of lymph node detection on the prognosis of pT1c33N0M0 NSCLC patients, ultimately aiming to improve postoperative survival rates (*Maniwa et al., 2020*). Due to the limitations of this study, the sample size included in this study is relatively small, and the follow-up time (5-year survival rate) of patients is relatively short, which has certain limitations to fully understand the impact of lymph node detection on prognosis. In the future, the sample size will be expanded and the follow-up time extended to obtain more comprehensive information on the impact of lymph node detection on the prognosis of patients with pT1c33N0M0 NSCLC.

### Funding

The authors received no funding for this work.

### Competing Interests

The authors declare there are no competing interests.

### Author Contributions

- Wei Yang conceived and designed the experiments, performed the experiments, analyzed the data, prepared figures and/or tables, authored or reviewed drafts of the article, and approved the final draft.
- Luyi Wang conceived and designed the experiments, performed the experiments, analyzed the data, prepared figures and/or tables, authored or reviewed drafts of the article, and approved the final draft.

### Human Ethics

The following information was supplied relating to ethical approvals (i.e., approving body and any reference numbers):

All samples obtained in this study were approved by the ethics committee of the Beijing Chest Hospital, Capital Medical University and abided by the ethical guidelines of the Declaration of Helsinki.

### Data Availability

The raw data is available in the Supplementary File.

### Supplemental Information

Supplemental information for this article can be found online at http://dx.doi.org/10.7717/peerj.16866#supplemental-information.

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
