# Peer review of "The prognostic significance of lymph nodes in patients with pT1c33N0M0 non-small cell lung cancer: a retrospective study"

_PeerJ, doi:10.7717/peerj.16866_

## Round 0.1 · original submission · Minor Revisions

Here are my comments:
1. The methods used for lymph node detection and the criteria for defining the N1 station are not clearly described. Please provide a detailed methodology, which is crucial for reproducibility and for other researchers to validate the findings.
2. This current study focuses on 5-year survival rates, but longer follow-up may be necessary to fully understand the prognostic impact of lymph node detection over time.
3. The use of the term "pT1c33N0M0" is standard? The authors should confirm the terminology to align with accepted staging nomenclature.
4. The authors should check that all the literature that appears in the body of the main text is correctly listed in the reference section.

Reviewer 1 ·

Basic reporting

The manuscript provides a comprehensive analysis of the correlation between the number of N1 station lymph node dissections and the prognosis of pT1c33N0M0 non-small cell lung cancer (NSCLC) patients. The study, conducted at a single center with a cohort of 255 patients, has identified that the quantity of identified lymph nodes serves as an independent prognostic determinant for these patients, with a minimum of three or more lymph nodes detected in the N1 station resulting in better long-term survival rates. The manuscript is clear and technically correct, with a detailed introduction and background that addresses the significance of accurate lymph node detection in NSCLC patients. The study effectively fills the knowledge gap by focusing on the specific impact of N1 station lymph node dissections on prognosis, which had been an area of limited investigation. The technical standard employed is robust, supported by detailed statistical methods and analysis, enabling replicability. The underlying data is robust, and the control measures are well-documented, ensuring the reliability of the findings. Overall, the manuscript presents a valuable contribution to the field of NSCLC research.

Experimental design

1. Describing the frequency and duration of follow-up visits would enhance the understanding of the study's methodology.

Validity of the findings

1. Adding a brief explanation of the potential clinical implications of the "Outcome Measures" section, including the significance of the 5-year survival rate, would enhance the reader's understanding.
2. The statistical significance of the findings related to the number of lymph nodes detected should be explained in greater detail within the results section.
3. The limitations of the study, such as the relatively small sample size, should be acknowledged in both the discussion and the conclusion.

Additional comments

1. The introduction should provide a brief overview of the current understanding of lymph node dissection in NSCLC and highlight the existing gaps in knowledge which the study aims to address.
2. It would be beneficial to provide a brief explanation in the discussion of how the findings align with or contribute to existing literature.
3. The discussion provides a good overview of the study's findings, but further discussion on how the results align with existing literature and implications for clinical practice would enhance the depth of the discussion.

Reviewer 2 ·

Basic reporting

The manuscript provides a retrospective analysis of 255 patients with pT1c33N0M0 NSCLC, investigating the correlation between the quantity of N1 station lymph node dissections and patient prognosis. The study identifies the minimum number of N1 station lymph nodes necessary for a better prognosis, highlighting the independent prognostic value of identified lymph nodes in pT1c33N0M0 NSCLC. However, the study has some limitations, including a single-center retrospective design, potential case selection bias, and a relatively small sample size. The manuscript also lacks further analysis on specific T stages, which affects the generalizability of the results. While the findings are intriguing and potentially impactful for clinical practice, the limitations undermine the robustness and applicability of the study. Therefore, the manuscript would benefit from a multicenter study with a larger, more diverse sample and a multidimensional approach to validate the impact of lymph node detection on the prognosis of pT1c33N0M0 NSCLC patients. Additionally, expanding the analysis to include specific T stages would enhance the clinical relevance of the findings.

Experimental design

a) The manuscript should include a discussion of the potential limitations of the study design, including any inherent biases associated with the retrospective analysis of patient data.

Validity of the findings

a) The significance levels for variable inclusion in multivariate analysis and ROC curve analysis seem unusually high (P < 0.3), and it would be beneficial to provide a rationale for this choice.
b) The manuscript should include a discussion of potential sources of error in the data collection process.

Additional comments

a) The introduction provides a comprehensive overview of the significance of the study on NSCLC. However, providing specific statistics related to the incidence and mortality rates would further enhance the context.
b) In the introduction, it would be beneficial to briefly introduce the significance of identifying an optimal number of N1 station lymph node dissections for pT1c33N0M0 NSCLC patients, as this would provide context for the study.
c) The information provided regarding the distribution of lymph nodes and patient demographics is comprehensive; however, it would be helpful to include more details on the patient demographics, such as age distribution and gender representation, to provide a more comprehensive overview of the study population.

Reviewer 3 ·

Basic reporting

No comment.

Experimental design

1-Provide more information about how to identify and classify lymph nodes.

Validity of the findings

2-Clarify the specific statistical tests used for comparing clinical and pathological data among patient groups
3-Results should expand on the significance of the findings related to the number of identified lymph nodes and their association with patient prognosis

Additional comments

4-Expand the introduction to include a brief overview of the importance of lymph node dissection in NSCLC treatment
5-Include specific statistical data regarding the incidence and mortality rates of NSCLC to support the statement of its high prevalence
6-The introduction provides a comprehensive overview of the significance of NSCLC and the study’s objectives. However, it could benefit from a brief explanation of the current approach to lymph node detection in NSCLC patients to provide context for the study.

---

## Round 0.2 · accepted · Accept

The reviewer's concerns and mine were well addressed. I think this revised version could be considered for publication in this journal.

Reviewer 1 ·

Basic reporting

The article has been revised and the structure has become clear, using professional English throughout.
The literature references provide sufficient on-site background.

Experimental design

After revision, the problem studied in the article is clearly defined, relevant, and meaningful. It illustrates how research can fill identified knowledge gaps.

Validity of the findings

The author has provided all basic data; They are robust, statistically reliable, and controllable.
The conclusion section is sufficiently stated and relevant to the original research question.

Additional comments

No comment.

Reviewer 2 ·

Basic reporting

I have reviewed the author's revised article and found it to be very well revised. The author has responded to my comments one by one. After revision, the content of the article has become coherent and clear, with charts and content related, and sufficient resolution.

Experimental design

I see that the author also provided sufficient information in the method description section so that other researchers can replicate it, the research question is clearly defined, and relevant. The modifications were made well and met the requirements for publication in the journal.

Validity of the findings

After modification, the conclusion section is presented appropriately and is relevant to the original issue under investigation, and is limited to issues supported by the results. The provided raw data is also reasonable.

Additional comments

I don't have any further comments, I believe the article has met the publishing standards.